# Validity and Reliability of the Insole3 Instrumented Shoe Insole for Ground Reaction Force Measurement during Walking and Running

**DOI:** 10.3390/s22062203

**Published:** 2022-03-11

**Authors:** Leora A. Cramer, Markus A. Wimmer, Philip Malloy, Joan A. O’Keefe, Christopher B. Knowlton, Christopher Ferrigno

**Affiliations:** 1Department of Orthopedic Surgery, Rush University Medical Center, 1611 W Harrison Street, Suite 201, Chicago, IL 60612, USA; leora_a_cramer@rush.edu (L.A.C.); markus_a_wimmer@rush.edu (M.A.W.); philip_malloy@rush.edu (P.M.); christopher_knowlton@rush.edu (C.B.K.); 2Department of Anatomy and Cell Biology, Rush University Medical Center, 600 S. Paulina St., Suite 507, Chicago, IL 60612, USA; joan_a_okeefe@rush.edu; 3Department of Physical Therapy, Arcadia University, 450 S. Easton Road, Glenside, PA 19038, USA

**Keywords:** gait mechanics, pressure insole, ground reaction force, validation, reliability

## Abstract

Pressure-detecting insoles such as the Insole3 have potential as a portable alternative for assessing vertical ground reaction force (vGRF) outside of specialized laboratories. This study evaluated whether the Insole3 is a valid and reliable alternative to force plates for measuring vGRF. Eleven healthy participants walked overground at slow and moderately paced speeds and ran at a moderate pace while collecting vGRF simultaneously from a force plate (3000 Hz) and Insole3 (100 Hz). Intraclass correlation coefficients (ICC) demonstrated excellent vGRF agreement between systems during both walking speeds for Peak 1, Peak 2, the valley between peaks, and the vGRF impulse (ICC > 0.941). There was excellent agreement during running for the single vGRF peak (ICC = 0.942) and impulse (ICC = 0.940). The insoles slightly underestimated vGRF peaks (−3.7% to 0.9% bias) and valleys (−2.2% to −1.8% bias), and slightly overestimated impulses (4.2% to 5.6% bias). Reliability between visits for all three activities was excellent (ICC > 0.970). The Insole3 is a valid and reliable alternative to traditional force plates for assessing vGRF during walking and running in healthy adults. The excellent ICC values during slow walking suggests that the Insole3 may be particularly suitable for older adults in clinical and home settings.

## 1. Introduction

Three-dimensional motion analysis is a valuable tool for quantifying gait parameters, monitoring disease progression, and guiding rehabilitative and medical treatment decisions [1,2]. Motion analysis quantifies ground reaction forces (GRF) using highly accurate force plates along with specialized cameras for capturing motion [2,3]. Such equipment is expensive and requires dedicated space and extensive training to use, thereby limiting such gait assessments to select centers and laboratories [4,5], and restricting their use in most clinical settings. Due to its small capture volume, many motion analysis setups can only detect a few stride sequences, and the force plate is limited to detecting the GRF for only a single step [5,6]. These limitations warrant exploration and development of portable, less expensive, and more user-friendly devices for quantifying GRF in real-world settings like clinics and home environments. Additionally, researchers and clinicians are often interested in the vertical component of the GRF (vGRF), particularly in studies where the force in other planes is lower in priority [6,7].

Viable alternatives to force plates for calculating an estimate of the vGRF include other technologies such as pressure mats [8] and an array of instrumented shoe insoles [8,9,10]. Of these alternatives, instrumented shoe insoles offer the most potential for use outside of a laboratory setting [11]. Until recently, sensor-enabled insoles were bulky, often containing external wiring harnesses, telemetry units, and power sources [8,12] that require extensive technical expertise and set-up time and potentially alter natural gait, making them impractical for clinical and home use [13]. Technical advances have enhanced accuracy and now allow for insoles to be wireless, allowing for force assessment that is more affordable, more time efficient, and less obtrusive to patients. One such device is the Insole3 (Moticon ReGo AG, Munich, Germany), seen in Figure 1. The Insole3 is a completely wireless insole powered by a coin cell battery. It stores data onto an onboard memory card while transmitting data via Bluetooth to a computer or smartphone. The Insole3 is flexible and incorporates sixteen capacitive pressure sensors, a three-dimensional accelerometer, and a three-dimensional gyroscope for measuring plantar pressure and motion. The insoles’ firmware integrates pressure measurements under the foot to allow for the continuous computation of vGRF estimates.

Such technologically advanced insoles greatly improve their utility in rehabilitation science and in conducting studies in ecologically valid environments including the home and community [13]. These features are especially useful for evaluative diagnostics, particularly in patients with abnormal force parameters contributing to disease progression, pain, morbidity, and reduced quality of life. Another novel feature of the Insole3 device is the addition of real-time audio feedback capability via Bluetooth connectivity to a smartphone, which can be used as a tool to assist in modifying gait patterns [12] and vGRF [14] in patients for whom excess force parameters are a concern, such as in those with musculoskeletal disorders. However, prior to being used for these applications, such an insole must be validated to confirm that it accurately estimates force.

The purpose of this study was to evaluate whether the Insole3 is a valid and reliable alternative to an in-ground laboratory force plate for measuring vGRF during slow walking, moderate-paced walking, and running. We hypothesized that there would be good to excellent agreement between the vertical force estimates from the insoles and the measured vGRF from the force plates, and good to excellent test–retest reliability of the insoles between two study visits.

## 2. Materials and Methods

### 2.1. Participants

This study was approved by the Institutional Review Board (Rush ORA-12021506). An a priori power analysis was performed using intraclass correlation coefficients (ICC) in the determination of reliability [15]. The power analysis yielded a sample size of 11 subjects with a power of 0.9, alpha of 0.05, a null assumption of ICC = 0, and an alternative hypothesis of ICC = 0.75 to signify good agreement [16]. Individuals who self-reported as healthy, pain-free, functional in daily activities without the need of assistance, and absent of history of surgery on the ankles, knees, and hips were recruited through local university networks. During informed consent, subjects were asked to return for a second study visit within one to four weeks following their first visit.

### 2.2. Motion Capture: Force Plates and Cameras

Two force plates (Bertec Corporation, Columbus, OH, USA) embedded in a walkway recorded GRF at a sampling rate of 3000 Hz. Twelve optoelectronic cameras (Qualisys, Gothenburg, Sweden) recorded the position data of reflective markers at 120 Hz during walking and 300 Hz during running. Five markers were placed bilaterally on the lateral and medial malleoli, and on the shoe surface overlying the fifth metatarsal tuberosity, second metatarsal head, and posterolateral heel as previously described [12]. Following standing reference trials, the medial markers were removed for dynamic trials. Video data were used to facilitate synchronizing the data from both systems and marker data were used to confirm that speeds fell within desired ranges.

### 2.3. Insoles

The Insole3 sampling rate was set to its maximum of 100 Hz. The pressure sensors have a mean resolution of 0.25 N/cm^2^ with a range of 0–50 N/cm^2^ as per manufacturer’s documentation [17]. The insoles connect wirelessly via Bluetooth to an Android smartphone containing the insole manufacturer’s OpenGo mobile application (Moticon ReGo AG, Munich, Germany). Insole data were recorded to the insole’s onboard SD memory card and subsequently transferred wirelessly to the OpenGo Software (Moticon ReGo AG, Munich, Germany) on a laptop computer following each visit.

The insoles were fitted into a standardized shoe (Adidas low-cup VRX, model DB3176) after removing the shoe’s original insole. The athletic shoe has a flat footbed to minimize interference with the insole’s contour while providing ample cushioning for walk and run trials. Subjects wore a standardized sock without seams on the plantar surface. Subjects participated in a familiarization period of at least 30 min that consisted of the subject sitting, standing, and walking in the standard shoes and insoles. This period also allowed the temperature of the sensors in the insoles to warm-up, limiting drift and associated measurement errors [18]. An automatic zeroing process, which occurs continuously once the phone application connects to the insoles, was done to reduce drift during recordings and errors associated with high mechanical impacts.

The insoles were calibrated to the subject’s bodyweight, as measured by a balance beam scale. The insoles’ built-in calibration procedure involved four motion tasks prompted by the application: slow walking while maintaining fluid motion for 40 s, standing still for 10 s, repeated anterior-posterior weight shifts for 10 s, and repeated side-to-side weight shifts for 10 s. Successful auto-zeroing was confirmed by a zero total force reading under a raised foot while standing on the contralateral limb.

### 2.4. Gait Conditions

Three gait conditions were evaluated in the following order: (1) moderate-paced walking, (2) slow walking, and (3) running. The following speeds for gait trials were targeted to minimize temporal variability: 1.2–1.4 m/s for moderate-paced walking, 0.8–1.0 m/s for slow walking, and 3.3–3.7 m/s for running. Practice trials were performed before each condition to reach the predefined target speeds which were estimated in real-time using a stopwatch over a predefined distance of 20 feet, or 6.096 m [19].

After practice trials, dynamic trials were recorded simultaneously by the insoles, force plate, and motion capture system. Subjects repeated gait trials until a minimum of five successful recordings for each of the left and right leg were obtained at each of the two walking speeds and for running. Therefore, a minimum of 30 total trials were recorded for each subject. A trial was considered successful if the subject’s speed was within the target speed range, as determined with the stopwatch, and if their step of interest hit completely on the force plate. Subjects were provided rest periods between conditions and by request.

### 2.5. Data Processing

Motion data from the foot markers were processed in Visual 3D (C-Motion, Inc., Germantown, MD, USA) to better evaluate the gait speed for each trial; these speeds were used in the test–retest reliability analysis to compare for differences between the two visits. In cases where extra trials per side were collected, the five trials closest to the midpoint of the target speed range were selected for final validity and reliability analyses.

Data processing and analysis were performed using a custom code for R (R Version 4.0.3 (2020), Statistical Computing, Vienna, Austria). A flash of light from the phone controlling the insole, which was in clear view of a video camera from the motion capture system, marked the end of the recording. To synchronize relative timestamps between the force plates and the insoles, a flash of light from a smartphone communicating wirelessly with the Insole3 was captured by the video camera, marking the end of the trial recording. No smoothing, filtering, or interpolation were performed on the force datasets.

The discrete variables of interest for the walking trials included the first and second local maxima, defined as Peak 1 and Peak 2 of the stance phase, which indicate the initial contact and propulsion characteristics of the vGRF. Additionally, the local stance phase minima, defined as the valley, of the vGRF, which represents weight acceptance and mid stance, were analyzed. Since running includes a single maximum of the vGRF during stance phase, only this value was analyzed for the running trials. Impulse was calculated for walking and running trials as the area under the vGRF–time series data waveform during the duration of stance using a composite trapezoid method [20].

### 2.6. Statistical Analyses

Data from the first visit only were used to assess overall agreement between the Insole3’s force-estimated data and the measured force data from the force plate. Insole3 data from both Visit 1 and Visit 2 were used to evaluate its test–retest reliability. For both the validity and reliability analyses, classic methods of ICC, confidence intervals (CI), and Bland–Altman plots [21,22] were used.

For the validity assessment, ICC (3,k) was used for a mean-rating (k = 2), absolute agreement, and two-way mixed effects model between subject and device. ICC statistics were calculated using the psych package in R. For interpreting the strength of the ICC values, the following categories suggested and described by Koo et al. were used: excellent (>0.90), good (0.75–0.90), moderate (0.50–0.75), or poor (<0.50) [16]. For the validity analysis, percent biases were calculated as the mean difference of measurements between the devices, as a proportion of the force plate values. Modified Bland–Altman plots with limits of agreement (LOA) display mean differences in vGRF between the two devices for each of the eleven subjects’ values. LOA were calculated as two standard deviations of differences between the Insole3 and force plate means.

Test–retest reliability was assessed using a mean-rating (k = 2), consistency, with a two-way mixed effects model between subjects and visits [16]. For the reliability analysis, percent biases were calculated as the mean difference of measurements between visits, as a proportion of the mean of Visits 1 and 2. Walking and running speeds between visits were compared for significant differences using a paired two-tailed t-test.

## 3. Results

### 3.1. Subjects

Eleven subjects participated in this study. Subject demographics are presented in Table 1.

### 3.2. Absolute Agreement between Insole3 and Force Plate

Mean speeds were 0.94 ± 0.04 m/s for the slow walk, 1.31 ± 0.04 m/s for the moderate-paced walk, and 3.72 ± 0.22 m/s for the run; running was thus performed at or slightly above the upper limit of the targeted range. Five of each left and right foot-strike per condition were included in the final analysis.

Descriptive statistics and ICCs for the validity comparison of the Insole3’s estimated forces with the measured forces of the force plate are shown in Table 2. With ICC values at or above 0.940 for Peak 1, Peak 2, valley, and impulse, during slow and moderate-paced walking speeds and running, the Insole3 vGRF estimates demonstrate excellent agreement with the force plate vGRF. The highest ICC values were seen in slow walking, with slightly lower but still robust ICCs for moderate-paced walking and running. F-tests indicated significant intraclass correlations between devices (*p* < 0.0001).

Modified Bland–Altman plots of mean differences between the insoles and force plate demonstrate that the majority of points lie within the LOA (Figure 2). The average of mean differences across all subjects for metrics in each condition provides insight into possible measurement bias between the two systems. Close inspection of the plots suggests that the insoles tended to slightly underestimate peak forces and valleys during walking conditions (% bias = −3.7–1.1%). Peak force during running was split between over- and underestimations and had a relatively small mean bias (% bias = 0.9%). For all three conditions, the Insole3 tended to overestimate impulses when compared with the force plates (% bias = 4.2–5.6%). Although not directly assessed, bias of the insole appears independent of vGRF magnitude during walking conditions, while there is potentially an inverted relationship between vGRF and insole bias during running trials.

Representative vGRF–time curves from the force plate and insole for both walking conditions demonstrate a characteristic double-peak, and the running condition in all 11 subjects demonstrates a characteristic single peak (Figure 3). In a single representative sample, an overlay of the waveforms visibly shows high agreement along the force–time continuum, especially during slow walking speeds. Also noticeable is a small time-related lag in the vGRF signal at the end of the stance phase (Figure 3, dotted ellipses). During running trials when the vGRF slope is steepest, a lower insole sampling rate results in a low number of data points particularly during weight acceptance. A review of the two-dimensional videos for each runner revealed that 10 of the 11 subjects in the study ran with a rearfoot landing pattern [23], as confirmed by review of the video camera recordings. Those with a rearfoot landing pattern showed an initial spike in their representative vGRF–time curve, termed the impact transient [24], from the force plate while such impact transients were absent for the midfoot striker and from all 11 representative vGRF–time curves from the Insole3.

### 3.3. Test–Retest Reliability

No significant differences in mean speed were found between visits as determined by paired two-tailed t-tests for the slow walks (*p* = 0.2506) and moderate-paced walks (*p* = 0.6760). Mean running speed has a *p*-value of 0.0738 between visits. Excellent test–retest reliability was found for all dependent variables, including Peak 1, Peak 2, valley, and impulse, for the Insole3 between Visits 1 and 2, and results were comparable with the force plate’s test–retest reliability (Table 3). For all dependent variables, mean test–retest ICC values for the insoles ranged from 0.970–0.996 and F-tests indicated significant non-zero intraclass correlations between visits (*p* < 0.0001).

## 4. Discussion

The results of this study demonstrate that the fully wireless and compact Insole3, with 16 pressure sensors, has excellent absolute agreement with the force plate for estimating peak vGRF and impulse during walking and running, and excellent test–retest reliability between the two study visits. The Bland–Altman plots and percent bias values suggest that the Insole3 tends to slightly underestimate the vGRF Peak 1, Peak 2, and the valley between peaks during the stance phase during both speeds of walking, which is similar to previously reported findings from studies using other sensor-enabled insoles similar to the Insole3 [9,25]. This underestimation may be expected for insole measurements as the viscoelastic properties of a rubber sole attenuate the transference of GRFs to the interface between the top of the shoe sole and the plantar sole of the foot. Peak forces during running were nearly evenly over- and underestimated by the Insole3, suggesting that when using the Insole3 for running, it is likely beneficial to take averages of several trials in order to obtain the most accurate vGRF assessment. Other insoles have been shown to underestimate running vGRF peaks [9,25,26].

Vertical force impulses calculated by the Insole3 tended to be slightly higher than those measured by the force plate during all three conditions, whereas previously tested sensor-enabled insoles tended to output lower impulses [9,25]. A contributing factor to the Insole3’s overestimations of impulse is a slight time-related lag in the vGRF signal found by the Insole3 compared with the force plate in the region between terminal stance and an offloaded state. The tendency of the Insole3 to slightly overestimate impulse is relatively minor, though somewhat consistent between subjects and trials; it should therefore be considered if relying on the insole’s force output during the latest parts of the stance.

The Insole3 had higher levels of agreement for vGRF peaks during walking (ICC = 0.941–0.986) than its predecessor, the OpenGo (ICC = 0.886) [6], and lower percent biases than the OpenGo in estimations of vGRF peaks and impulses during walking and running [25]. Excellent consistency was found for the Insole3 as well as the OpenGo, which was evaluated between trials rather than between visits [6].

Direct comparison of this study’s results with other concurrent validation studies is difficult due to different methodologies, but the results and a review of the literature suggest that the Insole3 has improved estimations of vGRF measurements compared with the OpenGo during walking and running, which may be due to hardware and firmware improvements. The increased number of pressure sensors from 13 in the OpenGo to 16 in the Insole3 provides better sensor coverage of foot area, and the increased sampling rate from 50 Hz to 100 Hz results in higher data resolution. Improvements in measurements when increasing the sampling frequency were also noted in the loadsol^®^ sensor insole when the recording rate was doubled from 100 Hz to 200 Hz [9]. The differences between the force plates’ sampling rate (3000 Hz) and the Insole3’s sampling rate (100 Hz) may account for some of the slight systematic biases in the insole’s estimations of vGRF [27]. This is most prevalent in the running curves seen in Figure 2, where the Insole3 was unable to record the impact transient of rearfoot landing during running. However, given its compactness, low cost, full integration, portability, and ease of use, our findings suggest that the Insole3 has great utility in most straight-lined gait applications where force measurements are needed outside of the laboratory.

Since a single insole size covers multiple shoe sizes, insole coverage underneath each foot will vary, as will the spatial distribution of the 16 sensors between insole sizes. Insole sizes tested in this study were limited by the availability of standardized footwear sizes. Study inclusion was limited to participants with shoe sizes between U.S. men’s/women’s sizes 6/7 through 12/13, restricting recruitment of otherwise eligible females with shoe sizes below US women’s 7. However, previous research suggests that there is little difference between males and females in their vGRF during running [28] and in their plantar pressures during walking [29]. Although few older subjects were recruited for this study, we evaluated slow walking conditions among participants. The highest ICC values and the lowest biases were obtained under those conditions, suggesting the Insole3′s applicability in older populations [30].

Potential sources of variance were reduced by standardized conditions, including standardized footwear and the use of pre-defined target speeds as opposed to self-selected paces. While the chosen gait speeds reflect normal self-selected paces of adults [31], and since vGRF parameters can vary with walking speeds [32,33], activities with higher ground-contact speeds such as fast walking, sprinting, cutting, or other agility movements could be further assessed for levels of agreement.

The high agreement of the Insole3 in estimating force during slow walking is promising for its use in older populations, since walking speeds progressively decline with aging and older populations demonstrate slow walking speeds similar to those used in the present study [30]. Motion analysis of older adults in a traditional motion analysis laboratory can be particularly difficult; a more portable and streamlined system would therefore be beneficial, especially considering the diminished functional capacity and decline in independence, general mobility, and transportation in many older adults. Assessing force in this population is important, especially in conditions related to osteoporosis, osteoarthritis, and post-traumatic fracture repair and hip replacement where partial weight bearing is often indicated and compliance is challenged [7].

## 5. Conclusions

This study demonstrates that the Insole3 is a valid and reliable alternative to the force plate for assessing discrete vGRF parameters during slow and moderate-paced walking, as well as moderately paced running in a healthy young adult population. These findings suggest that the Insole3 has applicability outside of specialized gait laboratories, such as in clinical and home settings. This insole may prove particularly beneficial in applications where subjects are bound to their homes and/or feedback about weight bearing is crucial, such as lower limb joint disease [11] and in the recovery periods following injury or surgery [34], or simply in the study of older adults who have difficulties travelling to specialized laboratories.

## Figures and Tables

**Figure 1 sensors-22-02203-f001:**
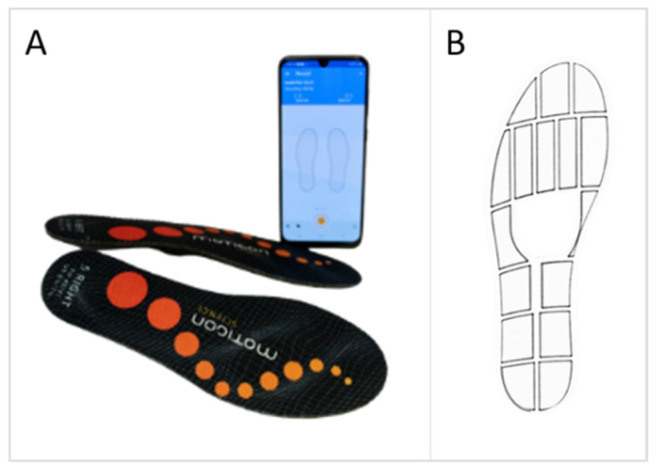
(**A**) The Insole3 (Moticon ReGo AG, Munich, Germany) and the OpenGo smartphone application. (**B**) Layout of the 16 pressure sensors and coordinate system. Figure 1B used and modified with permission from Moticon ReGo AG.

**Figure 2 sensors-22-02203-f002:**
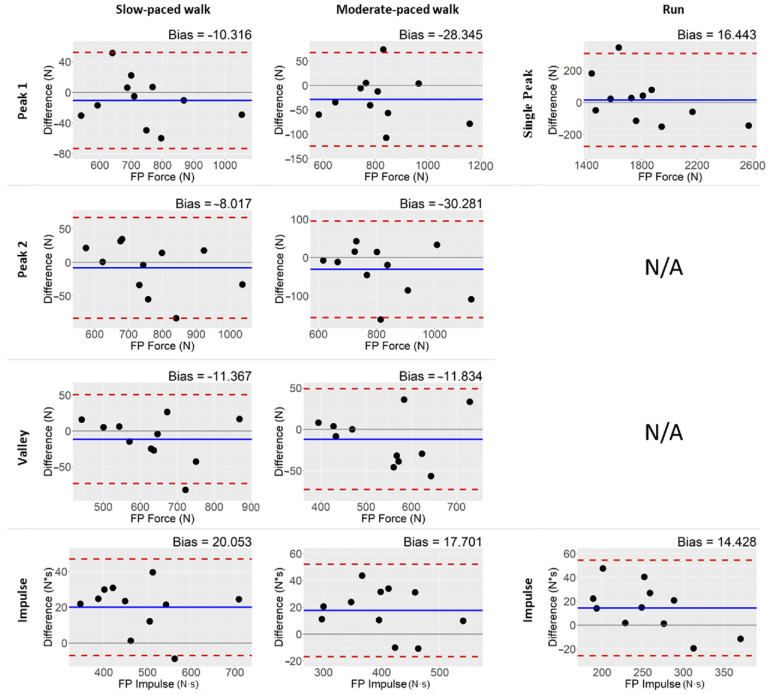
Modified Bland–Altman plots of Visit 1. The difference in mean force measures for each subject between the “gold standard” force plate and Insole3 is plotted. The mean bias is indicated by a solid blue line, and upper and lower limits of agreement are indicated by dashed red lines. Absolute biases are indicated at the top of each plot. Each point represents each subject (*n* = 11).

**Figure 3 sensors-22-02203-f003:**
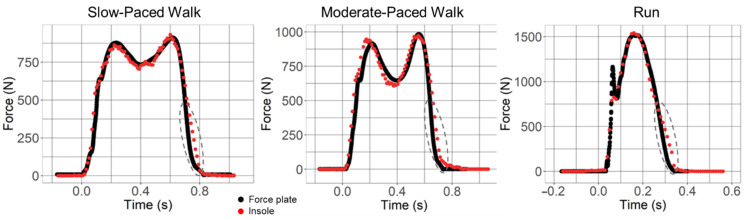
Representative vertical ground reaction force (vGRF) time plots from force plate (black) and Insole3 data (red) versus time. Both walks demonstrate characteristic double-peaked curves with two maxima (Peak 1 and Peak 2), and one minimum (valley), and the jogging trial represents a typical single-peaked curve with the presence of an impact transient prior to the peak. Target speed ranges were 1.2–1.4 m/s for the moderate-paced walk, 0.8–1.0 m/s for the slow walk, and 3.3–3.7 m/s for the jog. Gray dashed ellipses indicate regions that demonstrate a slight lag in the vGRF at the end of the stance phase with the insole.

**Table 1 sensors-22-02203-t001:** Subject characteristics (*n* = 11) and insole sizing. Each size of an Insole3 covers two European sizes (e.g., Size 4 covers European Sizes 38 and 39). The median European size was used for the insole size (e.g., for an individual wearing a Size 4 insole, their European insole size used was 42/43).

**Demographic**	**Mean ± Standard Deviation**
Number of Females (%)	3/11 (27%)
Height (inches)	68.5 ± 3.5
Age (years)	33.1 ± 16.7
Weight (kilograms)	74.2 ± 14.6
BMI (kilograms/m^2^)	24.6 ± 4.4
**Insoles**	**Median (range)**
Insole Size (European size)	42/43 (36/37–44/45)
Insole Size (Manufacturer Size)	Size 6 (3–7)

**Table 2 sensors-22-02203-t002:** Visit 1 agreement between the Insole3 and force plate for each vertical ground reaction force (vGRF) variable during slow walking, moderate-paced walking, and running trials. Absolute agreement between the devices is reported for two-way mixed effects and multiple measurements, with 95% confidence intervals (CI), F-test with *p*-value, and percent bias. ICC = Intraclass correlation coefficient.

Gait Condition	vGRF Variable	Force Plate	Insole	Absolute Agreement				
		Mean ± Standard Deviation	Mean ± Standard Deviation	ICC (3,k)	95% CI	F(10,10)	*p*-Value	Mean Bias (%)
Slow Walk	Peak 1 (N)	737.01 ± 139.91	726.69 ± 134.87	0.986	(0.960, 0.995)	72.766	0.0000	−1.4
	Peak 2 (N)	762.43 ± 133.45	754.42 ± 123.15	0.978	(0.938, 0.993)	46.067	0.0000	−1.1
	Valley (N)	634.95 ± 120.53	623.58 ± 115.80	0.981	(0.946, 0.994)	54.837	0.0000	−1.8
	Impulse (N·s)	481.71 ± 101.11	501.76 ± 99.36	0.986	(0.771, 0.996)	210.582	0.0000	4.2
Moderate-Paced Walk	Peak 1 (N)	815.70 ± 153.14	787.35 ± 155.32	0.968	(0.895, 0.990)	38.63	0.0000	−3.5
	Peak 2 (N)	818.13 ± 148.97	787.85 ± 139.37	0.941	(0.827, 0.980)	19.166	0.0000	−3.7
	Valley (N)	545.99 ± 103.76	534.16 ± 105.13	0.976	(0.931, 0.992)	44.094	0.0000	−2.2
	Impulse (N·s)	400.15 ± 72.23	417.85 ± 69.66	0.970	(0.803, 0.992)	64.278	0.0000	4.4
Run	Peak Max (N)	1814.98 ± 329.14	1831.42 ± 279.74	0.942	(0.836, 0.980)	17.383	0.0000	0.9
	Impulse (N·s)	256.29 ± 54.75	270.71 ± 44.96	0.940	(0.777, 0.981)	23.001	0.0000	5.6

**Table 3 sensors-22-02203-t003:** Test–retest reliability between visits for each vertical ground reaction force (vGRF) variable during slow walking, moderate-paced walking, and running. Consistency between Visit 1 and Visit 2 is reported for two-way mixed effects and multiple measurements; Visit 1 and Visit 2 means were compared for peak and valley forces (N) and impulses (N·s). ICC = Intraclass correlation coefficient; CI = Confidence interval.

Insole3				Test–Retest Consistency				
Condition	vGRF Variable	Visit 1 Mean ± SD	Visit 2 Mean ± SD	ICC(3,k)	95% CI	F(10,10)	*p*-Value	% Bias
Slow-paced walk	Peak 1 (N)	726.69 ± 134.87	727.24 ± 140.06	0.996	(0.987, 0.999)	224.49	0	0.1
	Peak 2 (N)	754.42 ± 123.15	745.05 ± 134.40	0.988	(0.963, 0.996)	81.05	0	−1.2
	Valley (N)	623.58 ± 115.80	631.70 ± 119.66	0.995	(0.985, 0.998)	193.61	0	1.3
	Impulse (N·s)	501.76 ± 99.36	510.66 ± 103.87	0.995	(0.984, 0.998)	182.40	0	1.8
Moderate-paced walk	Peak 1 (N)	787.35 ± 155.32	775.92 ± 161.95	0.983	(0.951, 0.994)	60.40	0	−1.5
	Peak 2 (N)	787.85 ± 139.37	779.08 ± 147.02	0.981	(0.943, 0.994)	52.55	0	−1.1
	Valley (N)	534.16 ± 105.13	540.71 ± 102.65	0.986	(0.959, 0.995)	72.17	0	1.2
	Impulse	417.85 ± 69.66	420.33 ± 79.53	0.983	(0.950, 0.994)	59.83	0	0.6
Run	Max vGRF (N)	1831.42 ± 279.74	1812.07 ± 282.82	0.970	(0.912, 0.990)	33.83	0	−1.1
	Impulse (N·s)	270.71 ± 44.96	266.02 ± 48.00	0.983	(0.950, 0.994)	59.39	0	−1.7
**Force Plate**								
Slow-paced walk	Peak 1 (N)	737.01 ± 139.91	735.55 ± 138.90	0.998	(0.994, 0.999)	491.40	0	−0.2
	Peak 2 (N)	762.43 ± 133.45	763.14 ± 135.32	0.999	(0.996, 0.999)	671.18	0	0.1
	Valley (N)	634.95 ± 120.53	642.20 ± 123.53	0.998	(0.993, 0.999)	403.99	0	1.1
	Impulse (N·s)	481.71 ± 101.11	492.60 ± 101.56	0.994	(0.984, 0.998)	180.85	0	2.2
Moderate-paced walk	Peak 1 (N)	815.70 ± 153.14	813.80 ± 160.77	0.990	(0.970, 0.997)	99.85	0	−0.2
	Peak 2 (N)	818.13 ± 148.97	817.67 ± 146.97	0.999	(0.996, 1.000)	765.10	0	−0.1
	Valley (N)	545.99 ± 103.76	550.67 ± 103.22	0.988	(0.965, 0.996)	84.69	0	0.9
	Impulse (N·s)	400.15 ± 72.23	405.25 ± 77.49	0.992	(0.975, 0.997)	119.70	0	1.3
Run	Peak vGRF (N)	1814.98 ± 329.14	1832.57 ± 336.15	0.992	(0.976, 0.997)	123.65	0	1
	Impulse (N·s)	256.29 ± 54.75	256.39 ± 52.55	0.991	(0.975, 0.997)	117.21	0	0.04

## Data Availability

The data presented in this study are available upon request from the corresponding author.

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
