# Peer review of "Validity and Reliability of the Insole3 Instrumented Shoe Insole for Ground Reaction Force Measurement during Walking and Running"

_sensors, 2022, doi:10.3390/s22062203_

Round 1

Reviewer 1 Report

lines 27-32: please fill in the data: ”Citation: Lastname, F.; Lastname, F.; Lastname, F. Title.” (on page 1, left)

lines 33-81 (pp. 1-2): please arrange the text with a Justified alignment (instead of the text aligned left, as it is now)

lines 83, 93, 103 etc.: please follow the MDPI guidelines concerning the layout of the subchapters

line 149: the phrase should end with a full stop.

lines 189-342: the same problem with the alignment

line 212: in the columns for ”Absolute agreement”, the ending parantheses for ICC(3,k) and F(10,10) appear on the next line; moreover, there is a blank space that should be eliminated: (F 10,10).

line 344: ”Author utions” - to be replaced, probably with ”Authors' contributions” or ”Contributions of each author”

Otherwise, overall I think that the contributions of the paper are clearly developed and presented. 

Reviewer 2 Report

In this paper, the authors evaluated whether the Insole3 is a valid and reliable alternative to force plates for measuring vGRF. Eleven healthy participants walked overground at slow and moderately-paced speeds and ran at a moderate pace while collecting vGRF simultaneously from a force plate (3,000 Hz) and Insole3 (100 Hz). The results indicate that Insole3 may be suitable for older adults in clinical and home settings. This article is clear, concise, and suitable for the scope of the journal. Several small suggestions are supplied:

1. Suggest the authors give more detail about the results in Fig.2.

2. Suggest the authors give more detail about the results about vertical ground reaction force (vGRF) time plots from force plate (black) and Insole3 data (red) versus during run.

3. Suggest authors improve the introduction part with other monitoring technologies such as:

Fiber Bragg Based Sensors for Foot Plantar Pressure Analysis 10.1007/978-3-030-29196-9_1
